# Rational Design of Novel Conjugated Terpolymers Based on Diketopyrrolopyrrole and Their Applications to Organic Thin-Film Transistors

**DOI:** 10.3390/polym15183803

**Published:** 2023-09-18

**Authors:** Shiwei Ren, Yubing Ding, Wenqing Zhang, Zhuoer Wang, Sichun Wang, Zhengran Yi

**Affiliations:** 1Zhuhai-Fudan Innovation Research Institute, Hengqin 519000, China; 2Key Laboratory of Organic Solids, Institute of Chemistry, Chinese Academy of Sciences, Beijing 100190, China; 3Key Laboratory of Colloid and Interface Chemistry of Ministry of Education School of Chemistry and Chemical Engineering, Shandong University, Jinan 250100, China; 4Laboratory of Molecular Materials and Devices, Department of Materials Science, Fudan University, Shanghai 200438, China

**Keywords:** polymer semiconductor materials, conjugated polymer, Stille coupling, OTFT device, terpolymer structures

## Abstract

Organic polymer semiconductor materials, due to their good chemical modifiability, can be easily tuned by rational molecular structure design to modulate their material properties, which, in turn, affects the device performance. Here, we designed and synthesized a series of materials based on terpolymer structures and applied them to organic thin-film transistor (OTFT) device applications. The four polymers, obtained by polymerization of three monomers relying on the Stille coupling reaction, shared comparable molecular weights, with the main structural difference being the ratio of the thiazole component to the fluorinated thiophene (Tz/FS). The conjugated polymers exhibited similar energy levels and thermal stability; however, their photochemical and crystalline properties were distinctly different, leading to significantly varied mobility behavior. Materials with a Tz/FS ratio of 50:50 showed the highest electron mobility, up to 0.69 cm^2^ V^−1^ s^−1^. Our investigation reveals the fundamental relationship between the structure and properties of materials and provides a basis for the design of semiconductor materials with higher carrier mobility.

## 1. Introduction

Research on polymers with conjugated architectures with alternating single and double bonds has attracted much attention because they are important candidate semiconductors for organic thin-film transistors (OTFTs) [1,2]. Device performance can be significantly enhanced by improving the molecular structure of organic semiconductors, and materials with carrier mobility in the range of 0.1–1 cm^2^ V^−1^ s^−1^ have potential applications in the fabrication of lightweight and flexible optoelectronic devices such as radio-frequency electronic trademarks, smart cards, sensors, logic circuits, and e-paper [3,4,5]. Organic semiconductor materials are generally classified into three categories: N-type, P-type, and ambipolar materials, according to the carrier species. N-type materials are mainly responsible for electron transport, and their research progress lags far behind that of P-type materials, which are responsible for hole transport [6,7,8]. The research on N-type materials is mainly constrained by the lack of diversity of acceptor species, and most of the selective acceptor species are concentrated in the systems of lactam structures, such as diketopyrrolopyrrole (DPP), isoindigo (IID), naphthalene diimide (NDI), perylene diimide (PDI), and so on [9,10]. The acceptor units affect the lowest unoccupied molecular orbital (LUMO) energy level of the materials, which determines their ability to attract electrons. Based on the failure to effectively develop new potential acceptor structural units, it is extremely important to modify the existing structures. From the perspective of structure-determined properties, the introduction of functional groups with strong electron-withdrawing ability is the most direct way to lower the LUMO energy level of the material. Some typical strong electron-withdrawing groups include fluorine atoms (F), chlorine atoms, bromine atoms, carbonyl groups, cyano groups, dicyanomethylene groups, nitrogen atoms, and so on [5,11,12]. On the other hand, increasing the proportion of acceptors in the molecular structure is an effective way to increase the electron mobility. Chen et al. summarized the two-acceptor strategy and three-acceptor strategy, which involve further insertion of acceptor units within the polymer system based on the D-A structure in order to form a D-A-A or D-A-A-A structure, respectively [13]. Recently, we reported the significant effect of the all-acceptor strategy on the improvement of electron mobility, further enriching the development and structural design strategies of N-type materials [14]. In addition to this, the research on terpolymers is receiving increasing attention because they can achieve flexible adjustment of energy levels by modulating the ratio of the components, and because their synthesis does not require much effort [15,16,17].

Here, we chose to study a polymer based on DPP combined with thiophene, which was first reported in 2015 [18]. The polymer shows a moderate hole mobility of 0.09 cm^2^ V^−1^ s^−1^ and is a P-type unipolar material that does not display electron transport properties (Figure 1). In order to improve its electron mobility, we envisaged the introduction of a strong electron-absorbing group F on thiophene, namely, fluorinated thiophene (FS). The F atom features a small atomic radius and a strong electron-accepting ability, which is an outstanding advantage for lowering the LUMO energy level of the material and not affecting the planarity of the whole molecule [19,20]. At the same time, the hydrophobicity of the F-containing polymer prevents moisture and oxygen from diffusing into the film, thereby improving the stability of the device in air. Furthermore, a strong electron-withdrawing unit, thiazole–thiazole (Tz), is introduced into the two-component polymers, and it is used as the third component to form the new ternary polymers [21,22]. In order to fully investigate the maximum electron mobility of this series of materials, four polymer materials with FS/Tz ratios of 100:0, 75:25, 50:50, and 25:75 were designed and prepared sequentially.

## 2. Materials and Methods

Materials: 3,6-di(thiophen-2-yl)-2,5-dihydropyrrolo[3,4-c]pyrrole-1,4-dione (DPP-S), (3,4-difluorothiophene-2,5-diyl)bis(trimethylstannane) (FS-Sn), N-bromosuccinimide (NBS), 5,5′-bis(trimethylstannyl)-2,2′-bithiazole (Tz-Sn), potassium carbonate (K_2_CO_3_), Pd catalysts, and organic solvents (e.g., methanol, chloroform (CHCl_3_), dimethylformamide (DMF), petroleum ether (PE), dichloromethane (DCM), etc.) were purchased from Derton Optoelectronic Materials Science Technology Co., Ltd. (Shenzhen, China) and Sigma-Aldrich (St. Louis, MI, USA), and they were used as received. DPP-S was synthesized as described in the previous literature [23]. Synthesis of DPP-S-C_8_C_10_ monomer: Argon was passed into a solution of DPP-S (3.50 g, 11.67 mmol) in anhydrous DMF (50 mL), and K_2_CO_3_ (4.83 g, 35.00 mmol, 3.00 eq) was added in batches over ten minutes. The mixture was slowly heated to 80 °C, and 9-(bromomethyl)nonane (9.25 g, 25.67 mmol, 2.20 eq) was added dropwise to the flask using a syringe. The mixture was stirred at 100 °C for 12 h. The mixture was then extracted with DCM, washed with water and brine, and dried with Na_2_SO_4_. After removal of the solvent under reduced pressure, the residue was purified by silica gel chromatography by eluate (PE:DCM = 4:1) to afford an orange–red powder (6.13 g, 61.0%); δ ^1^H NMR (400 MHz, chloroform-d) δ 8.88 (d, J = 3.9 Hz, 1H), 7.62 (d, J = 5.0 Hz, 1H), 7.27 (t, J = 5.7 Hz, 1H), 4.01 (d, J = 7.6 Hz, 2H), 1.95–1.86 (m, 1H), 1.35–1.12 (m, 32H), 0.92–0.83 (m, 6H); ^13^C NMR (100 MHz, chloroform-d) δ 161.78, 140.46, 135.25, 130.48, 129.88, 128.42, 107.98, 46.25, 37.77, 31.95, 31.91, 31.22, 31.21, 30.05, 30.04, 29.68, 29.66, 29.59, 29.53, 29.39, 29.33, 26.24, 22.72, 22.70, 14.16, 14.15 (Appendix A). Mass for C_54_H_89_N_2_O_2_S_2_^+^: 861.6360; found: 861.6359 (Appendix A). Synthesis of DPP-S-Br monomer: To a solution of DPP-S-C_8_C_10_ (2.50 g, 2.90 mmol) in chloroform (25 mL), we passed argon and then added NBS (1.08 g, 6.09 mmol, 2.10 eq). The reaction was completed by stirring the mixture at 60 °C for 0.5 h. The mixture was then extracted with DCM. After removal of the solvent under reduced pressure, the residue was purified by silica gel chromatography with the eluent (PE: DCM = 5:1) to give an orange–red powder (2.80 g, 95.2%); ^1^H NMR (400 MHz, CDCl_3_) δ 8.62 (d, J = 4.2 Hz, 1H), 7.21 (d, J = 4.2 Hz, 1H), 3.92 (d, J = 7.7 Hz, 2H), 1.94–1.81 (m, J = 7.2, 6.7 Hz, 1H), 1.38–1.13 (m, 32H), 0.91–0.85 (m, 6H); ^13^C NMR (75 MHz, CDCl_3_) δ 161.41, 139.41, 135.31, 131.43, 131.19, 118.95, 108.04, 46.36, 37.77, 31.93, 31.89, 31.19, 29.99, 29.65, 29.56, 29.50, 29.37, 29.29, 26.19, 22.70, 22.67, 14.12 (Appendix A). Mass for C_54_H_86_Br_2_N_2_O_2_S_2_: 1016.4497; found: 1016.4487 (Appendix A). Synthesis of PDPP−S−FS_n_−Tz_m_ polymer: Tz 50%: DPP-S-Br (200 mg, 196.23 µmol), FS-Sn (43.7 mg, 98.11 µmol 0.5 eq), Tz-Sn (48.5 mg, 98.11 µmol 0.5 eq), and triethyl phosphite (P(o-ty)_3_, 4.80 mg, 15.69 µmol) were dissolved in dry chlorobenzene (10 mL). The Schlenk tube was vented with argon over ten minutes, after which tris(dibenzylideneacetone)dipalladium ([Pd_2_(dba)_3_] catalyst, 3.6 mg, 3.92 µmol) was quickly added. The polymerization reaction was stirred at 130 °C for three days and gradually returned to room temperature. The purification was carried out by Soxhlet extraction with hexane (12 h), methanol (12 h), ethyl acetate (8 h), and acetone (10 h). Finally, the target polymer was obtained using chloroform phase. The fractions were evaporated and concentrated, precipitated into methanol (200 mL), and filtered to give the polymeric material in the form of a dark powder, which was dried under vacuum for 5 h (80 °C) to give Tz 50% (199.4 mg, 88.8%). The other three polymerizations were synthesized under similar reaction conditions and showed similar yields, except that the ratio of the two monomers varied.

Characterization: Nuclear magnetic resonance spectra of small molecules and polymers were measured in deuterated chloroform (NMR, Bruker AVANCE 400, Mannheim, Germany). Solid-state NMR: ^13^C cross-polarization magic angle spinning nuclear magnetic resonance (^13^C CP/MAS NMR) spectra were recorded on a Bruker Avance III 400MHz spectrometer (Germany). Samples were packed in 4 mm ZrO_2_ rotors, which were spun at 8 kHz in a double-resonance MAS probe. Mass spectrometry analyses were performed in ESI mode on a magnetic resonance mass spectrometer (solariX, Memmingen, Germany). The molecular weight of the polymer was estimated by high-temperature gel permeation chromatography (Agilent PL-GPC 220, Santa Clara, CA, USA). The solvent flow phase chosen here was 1,3,5-trichlorobenzene to fully ensure the solubility of the material. Thermal stability measurements were performed on a thermal analysis system in a nitrogen atmosphere with a heating rate of 10 °C min^−1^ (HITACHI STA200, Tokyo, Japan). Elemental analysis was conducted using an organic elemental analyzer (UNICUBE-Elementar, Langenselbold, Germany) in CHNS mode. The Fourier-transform infrared (FT-IR) spectra were collected using an FT-IR spectrometer (VERTEX 70v, Bruker, Germany) in a vacuum. Photochemical analyses were conducted using a UV–vis spectrometer (UV3600i, Shimadzu, Japan) using the standard quartz cell and the quartz chip (15 mm × 15 mm × 1 mm). The concentration of the solution was approximately 0.5 mg/mL, and the solvent was anhydrous chlorobenzene. The polymer semiconductor solution was spin-coated onto pre-cleaned quartz plates, followed by thermal annealing at 150 °C for ten minutes. Electrochemical measurements were performed under an argon atmosphere using cyclic voltammetry in acetonitrile solutions containing tetrabutylammonium hexafluorophosphate (TBAPF_6_) (CHI760E, CH Instruments, Bee Cave, TX, USA) at a scan rate of 50 mV/s. An Ag/AgCl electrode, platinum electrodes, and glassy carbon electrodes were used as reference, counter, and working electrodes, respectively. Then, 5 µL of a solution of the anhydrous chlorobenzene of the polymer, in the concentration range of 0.5 mg/mL to 1 mg/mL, was weighed sequentially with a pipette gun, dropped onto a glassy carbon electrode, and allowed to evaporate naturally at room temperature (the diameter of the circular hole of the electrode used was 4 mm).

OTFT device: Highly doped N-type silicon (Si) wafers with a silicon dioxide (SiO_2_) content of 300 nm were used as substrates, which were ultrasonically cleaned in deionized water, acetone, and isopropanol for 5 min, sequentially, and then processed in the UV zone for 25 min. After that, the substrate was modified with octadecyltrichlorosilane (OTS). Specifically, 1.3 µL of OTS and 1 mL of trichloroethylene were thoroughly mixed and then spin-coated on the substrate at 3000 rpm for 30 s. The substrate was then placed in a glass desiccator and dried under an atmosphere of NH_3_ for 8 h. For the preparation of semiconductor films, the conjugated polymers were pre-dissolved in chlorobenzene solvent (concentration 5 mg/mL) and heated overnight at 80 °C with stirring. Subsequently, the polymer solution was spin-coated onto OTS-treated Si wafers at 2000 rpm for 1 min, and then annealed at 200 °C for 30 min in a glove box (nitrogen atmosphere). Finally, source/drain electrodes of Au were deposited by evaporation in a vacuum (W/L = 5, W = 1000 µm, L = 200 µm) to finish the device preparation. The electrical performance of the OTFT device was assessed using a B1500A semiconductor parameter analyzer (Keithley, Cleveland, OH, USA) in a N_2_ glovebox. 

GIWAXS: The grazing-incidence wide-angle X-ray scattering images were acquired on beamline BL1W1A of the Beijing Synchrotron Radiation Facility. An Eiger detector was employed, and the pixel was 0.075 mm. The beam center and the sample-to-detector distance were calibrated with LaB6. The monochromatic wavelength of the light source was 1.5406 Å, the photon energy was 8.05 keV, and the grazing-incidence angle was 0.2°. The exposure time of the samples was 30 s. 

AFM: Atomic force microscopy images were obtained using an SPA300HV from Seiko Instruments Inc. (Chiba, Japan) in tapping mode (scanning range 5 × 5 µm). The scanning probe was performed using the OLTESPA-R3 silicon cantilever (Bruker, Germany). The elastic coefficient of the probe was 2 N/m. 

## 3. Results

### 3.1. Synthesis and Structural Analysis

The synthesis route of the series of polymers PDPP−S−FS_n_−Tz_m_ obtained based on Stille coupling polymerization is shown in Figure 2. DPP-S was synthesized with good yields using commercially available 2-cyanothiophene as a precursor. In order to improve the solubility of DPP-S, its N position was chemically modified to introduce an alkyl chain. Then, 9-(bromomethyl)nonadecane was introduced under base reaction conditions relying on an alkylation reaction to produce DPP-S-C_8_-C_10_, which possessed excellent solubility in common organic solvents such as n-hexane, chloroform, and ethyl acetate. Br atoms can be easily introduced on thiophene using NBS as a reagent, and the resulting DPP-S-Br can be used as a monomer for Stille coupling. The other two Sn-containing monomers, Tz-Sn and FT-Sn, were polymerized with DPP-S-Br under palladium catalysis. The Stille coupling reaction was carried out over a period of 72 h to ensure that the target products were polymerized to a sufficient extent. Chlorobenzene was chosen as the solvent due to its high boiling point and its excellent solubility for the polymers. The molecular weight of the polymers increased as the polymerization process progressed, resulting in a corresponding decrease in solubility in other common organic solvents. In order to ensure that the molecular weights of the four polymers were within a certain range, we strictly controlled the reaction time and the chemical equivalent ratios, including the amounts of catalyst and ligand. The polymers were purified using the Soxhlet extraction technique, and the successive use of hexane, acetone, methanol, and ethyl acetate effectively removed oligomers and small molecules from the reaction system. Polymers or oligomers with shorter chain lengths dissolve in acetone or hexane and exhibit a blue–violet color. The target polymers with longer chain lengths dissolve in chloroform and appear as dark green. 

The weight-average molecular weight (Mw) and number-average molecular weight (Mn) of the four polymers were in the region of 100.0 kDa and 25.0 kDa, respectively, as measured by high-temperature gel permeation chromatography (Appendix A). The relatively uniform distribution of Mn confirmed the effect of a suitable catalyst-to-ligand ratio and reaction time on polymer purity. The polymer dispersion index (PDI) calculated from the ratio of Mw to Mn is shown in Table 1 below. Based on the Mn values, it is roughly estimated that 21–25 repeating units were polymerized for each main chain. Elemental analyses of the four elements C, H, N, and O contained in the polymer Tz 0%, with theoretical values of 71.20%, 9.10%, 2.65%, and 9.90%, respectively, show that the contents of all four elements deviated from the theoretical values by less than 1% (Table 1). The relative increase in the proportions of N and S with increasing Tz content further indicated that the materials were sufficiently pure to be free of catalyst and ligand components. Liquid NMR analyses of the four polymers were difficult due to the poor quality of their spectra. On the one hand, this was due to their poor solubility, and on the other hand, it was due to the inability to clearly observe microscopic variations in the proportions of the components (Appendix A). The series of polymers were further characterized by NMR spectroscopy with solid-state NMR (Appendix A). The ^13^C NMR spectra were divided into an aromatic part and an alkyl chain part and corresponded well to the molecular composition of the polymers. The carbonyl group appeared near 160 ppm and, together with the typical peak of the central carbon peak of the thiazole, formed the leftmost broad single peak of low field in the spectrum. In addition, the four polymers were characterized by FT-IR spectroscopy, the results of which are shown in Appendix A. The typical carbonyl peaks of the polymers all appear near 1668 cm^−1^.

To test the thermal stability of this series of polymers, thermogravimetric (TGA) tests were performed under nitrogen, as shown below. All four polymers underwent significant decomposition up to 383 °C, with a 10% weight loss at 409 °C (Appendix A). There were no significant differences in overall thermal stability, except for a linear relationship at 70% weight loss, where the higher the Tz component, the more difficult it was to decompose completely. The polymers exhibited very poor solubility in non-chlorinated solvents such as ethanol, hexane, and tetrahydrofuran. Good solubility was only found in chlorinated solvents such as dichloromethane, chloroform, chlorobenzene, and trichlorobenzene at room temperature, and the solubility was further improved by heating to 50 °C. The dissolubility of the polymers in chloroform and chlorobenzene at room temperature was approximately 8 mg/mL and 15 mg/mL, respectively, and chlorobenzene was chosen as the solvent for subsequent characterization measurements and device preparation.

### 3.2. Photochemical Properties

The UV–vis absorption spectra of the polymer system in solution and film are shown below in Figure 3, and the corresponding spectral data are summarized in Table 2. Two absorption bands in the range of 350–500 nm and 600–900 nm can be observed for the polymers in solution. The high-energy absorption band located in the short-wavelength band can be attributed to the π–π* transition (band I), while the low-energy absorption band belongs to the intramolecular charge-transfer transition (band II). With the increase in Tz content, the main absorption peak (λ_max_) of absorption band I was significantly redshifted, from 406 nm to 444 nm (Figure 3a). The main absorption peak of absorption band II was blueshifted with increasing Tz content, and its corresponding λ_max_ was 906 nm, 899 nm, 895 nm, and 765 nm, respectively. The two polymers Tz 0% and Tz 75% exhibited distinct 0–0 and 0–1 peaks with shoulders at 767 nm and 701 nm, respectively. In contrast, the Tz 25% and Tz 50% polymers presented relatively broad absorption bands, which may be related to their reduced coplanarity due to better solubility. There was a redshift of close to 10 nm in the main absorption peak in band II from solution to film (Figure 3b). For example, the absorption of the Tz 75% polymer increased from 762 nm to 772 nm. These changes are related to the coplanar structure and better solid-state stacking with π–π stacking distances between the polymers. The optical bandgap of the polymer in solution (E_g_^soln^) was nearly 1.38 eV, and in the thin-film state the bandgap (E_g_^film^) was reduced to approximately 1.35 eV, as ascertained at the onset of the UV–visible absorption at 900 nm and 920 nm, respectively. The UV measurement diagrams using chloroform as the solvent also show dual absorption bands (Appendix A), and the data for the specific maximum absorption peaks are presented in Appendix A. The UV absorption spectra of the dimers of the polymers calculated on the basis of the DFT again show two absorption bands, the results of which are shown in Appendix A.

### 3.3. Electrochemical Properties

The redox potentials of the polymers of the PDPP−S−FS_n_−Tz_m_ series were investigated using cyclic voltammetry (CV). Their corresponding electrochemical data are summarized in Table 3. The differences in the LUMO energy levels calculated from the onset reduction potential for the four polymers were very small, in the vicinity of −3.68 eV (Figure 4). The extremely low-lying LUMO energy levels of the polymers are related to the presence of three strong electron-withdrawing components in the polymer backbone. On the other hand, it can be seen that by changing the proportion of F atoms or thiazoles in the molecular architecture, the effect on the LUMO energy level of the material is insignificant. In addition, the difference between their reduction peaks is also small, and the electrochemical reduction properties of the four polymers are similar. The polymers show obvious reversible cyclic reduction characteristics, and their reduction peaks are much more obvious than the oxidation peaks, indicating that their electron mobility is higher than their hole mobility. It is worth mentioning that the HOMO energy level of the polymer Tz 0% was significantly different from that of the other three polymers. Tz 0% possessed the highest HOMO energy level of −3.66 eV, which at the same time led to a minimum energy gap of merely 1.04 eV. The energy gap of the polymers was in the vicinity of 1.34 eV, which is close to their optical bandgaps. 

### 3.4. OTFT Performance

In order to characterize the charge transport behavior of these materials, we fabricated bottom-gate top-contact (BGTC)-structured OTFT devices based on PDPP-S-FS_n_-Tz_m_, and the corresponding device configurations are shown in Figure 5a. Considering that the HOMO level of the polymer matched that of the gold electrode (Au, ~−5.00 eV), Au was selected as the contact electrode. The field-effect mobility (µ) was calculated from the following equation:I_D_ = (W/2L) µC_i_ (V_GS_ − V_TH_)^2^

where I_D_ is the drain current, C_i_ is the capacitance of 300 nm silicon dioxide (Ci = 10 nF cm^−2^), W and L are the width and length of the channel, respectively, and V_GS_ and V_TH_ are the gate voltage and the threshold voltage, respectively. The average mobility values were obtained based on measurements performed on eight devices.

Table 4 summarizes the electron and hole mobility extracted from the transfer characteristic curve. It is evident that the OTFT devices based on PDPP-S-FS_n_-Tz_m_ have N-type dominant ambipolar transport characteristics, with electron mobility one order of magnitude higher than hole mobility (Figure 5b,d,f,h). The maximum electron mobility value in the Tz 0% polymer containing a thiazole-free system was only 0.38 cm^2^ V^−1^ s^−1^. The electron mobility improved with the increase in the thiazole ratio. Carrier mobility as high as 0.69 cm^2^ V^−1^ s^−1^ was shown at a thiazole–fluorothiophene ratio of 50:50. Continuing to increase the proportion of thiazole to 75% did not cause a further increase in mobility, which dropped to 0.57 cm^2^ V^−1^ s^−1^. On the other hand, the increase in electron mobility was inversely proportional to its hole mobility. The polymer Tz 50% exhibited the highest electron mobility while having the lowest hole mobility among the four materials. The Tz 0% showed the highest hole mobility of 0.07 cm^2^ V^−1^ s^−1^, and its average hole mobility was in the vicinity of 0.05 cm^2^ V^−1^ s^−1^. The output behaviors of the polymer-based devices are shown in Figure 5c,e,g,i, respectively.

### 3.5. Crystallinity Analysis of Polymer Films

We investigated the relationship between the crystallinity of polymer films and the electrical performance of OTFT devices by grazing-incidence wide-angle X-ray scattering (GIWAXS). Figure 6a–d show 2D-GIWAXS images of four polymer films prepared by spin-coating, and Figure 6e shows the corresponding in-plane and out-of-plane 1D profiles based on 2D-GIWAXS of the annealed films. The polymers all showed distinct out-of-plane (100) and in-plane (010) peaks, indicating that their films were predominantly edge-on in orientation. The polymer Tz 0% showed only (100) in the out-of-plane direction, while the other three polymers showed enhanced (100) diffraction intensity in the out-of-plane direction, accompanied by the appearance of secondary diffraction (200), which fully explains their enhanced crystallinity. This also explains the fact that the electron mobility of Tz 0% was the worst among the four. On the other hand, the presence of in-plane (100) diffraction for Tz 0% and Tz 25% indicated the gradual appearance of a face-on orientation. The stacking pattern of face-on crossed with edge-on is unfavorable for electron transport. The layered side-chain distances of Tz 0% and Tz 25% were 22.43 Å and 21.65 Å, respectively, which are loose compared to the stacking of Tz 50% and Tz 75% (Table 5). Tz 50% showed the shortest π–π stacking distance, which was calculated based on the in-plane (010) diffraction peaks to be only 3.95 Å. Tight stacking facilitates carrier transport and helps to achieve high mobility in OTFT devices, which explains why Tz 50% was the best performer of the four.

### 3.6. Morphological Analysis of Polymer Films

The morphology of the polymer semiconductor films was investigated by atomic force microscopy (AFM), and Appendix A shows the film height diagrams of the four polymer materials after annealing at 200 °C. Overall, the surface morphology of the polymer films shows a clear uniform distribution. The Tz 50% polymer film shows a dense fibrous structure, and the ordered interchain stacking in the fibers facilitates the charge transport. In contrast, the fibrillar shape of Tz 0% and Tz 25% gradually disappears into disordered aggregates. These results are consistent with the GIWAXS results, which are unfavorable for the electrical properties of the polymeric materials as the crystallinity becomes weaker.

## 4. Discussion

Compared to the hole mobility of the polymer PDPP-T based on DPP and thiophene, with moderate hole mobility, which was first reported in 2015, this work significantly improved the electron mobility of this type of material (Table 6). This was firstly related to the introduction of fluorinated thiophene, where the strongly electronegative F atom significantly reduced the frontline orbital energy level of the material. Bura et al. reported that DPP-based polymers prepared by introducing F atoms on the thiophene ring showed an electron mobility of 0.51 cm^2^ V^−1^ s^−1^ [24]. By replacing the electron donor thiophene with a benzene ring and further introducing a fluorine atom, the polymer DPP-F_1_Ph showed moderate electron mobility and a balanced bipolar character [25]. In addition to fluorine atoms, chlorine atoms are also promising electron-withdrawing moieties. Geng’s group reported that the polymer DPPTh-4Cl, prepared by introducing four Cl atoms to a thiophene ring, showed a high mobility of 0.81 cm^2^ V^−1^ s^−1^, which was attributed by the authors to the structural coplanarity caused by a weak Cl...S interaction and very low LUMO energy levels [26]. Other strong electron-attracting groups, such as cyano groups, can also be introduced on top of the F atom. Zhang et al. reported last year that the polymer F2CNTVT-DPP, which introduces −F and −CN on (E)-1,2-di(thiophen-2-yl)ethene (TVT), showed a maximum mobility of 2.03 cm^2^ V^−1^ s^−1^ [27]. On the other hand, this research is also related to simple terpolymerization to improve mobility. Similar work was reported by Chen et al. The polymer P2DPP-2FBTz, based on two DPPs and another receptor as a three-component composition, showed an average electron mobility of 0.61 cm^2^ V^−1^ s^−1^ [28]. Yi et al. reported DPP-2T-DPP-TBT, a bipolar material with high electron mobility, which was also obtained using three-component copolymerization [29]. In addition to the classical acceptor units used to prepare N-type materials, more and more research is focusing on the creation of new structures to improve the electron mobility of polymers. The polymer PFIDTO-T, with a ladder-like aromatic diketone structure, demonstrates highly promising electron mobility [30]. The polymer PBN-27, constructed with a bis-B-N-bridged bipyridine unit and a benzobithiazole unit, was developed and showed moderate N-type electron mobility [31]. Recently, the polymer PNFFN, constructed from IID-based isomers developed by Yu et al., showed the highest mobility of 1.82 cm^2^ V^−1^ s^−1^, which was further increased to 10.74 cm^2^ V^−1^ s^−1^ when poly-F atoms were introduced to the system [32]. Guo et al. reported an amide-based macro-heterocyclic polymer PDTzNTI, consisting of sulfur and nitrogen atoms, with unipolar N-type transport and significant electron mobility [33].

## 5. Conclusions

To summarize, we report in this work the changes in carrier mobility caused by the continued introduction of new acceptor units into two-component acceptor materials. The synthesis of these four polymers was achieved via Stille coupling polymerization, and proper control of the reaction and purification conditions allowed for the preparation of a series of polymers with high molecular weights and a relatively homogeneous molecular weight distribution. The polymers showed N-dominant properties and the highest electron mobility of 0.69 cm^2^ V^−1^ s^−1^ at a half–half ratio of thiazole to fluorothiophene, and the increase in mobility was mainly related to the close intramolecular Π-distance and good crystallinity, as shown by GIWAXS and film morphology analysis. Further increasing the proportion of thiazole failed to improve the electron mobility of the material in OTFT devices. Therefore, sophisticated molecular structure design strategies and component ratio modulation are key factors in controlling the material properties and device performance. Further development of N-type unipolar and ambipolar organic semiconductor polymer materials and enhancement of electron mobility are still in progress.

## Figures and Tables

**Figure 1 polymers-15-03803-f001:**
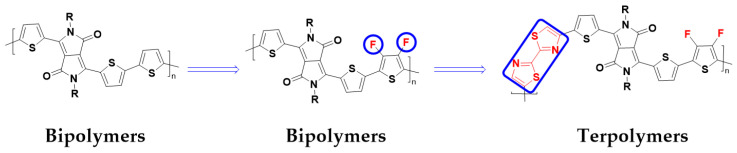
Molecular structure design strategies and chemical structure schematics from bipolymer to terpolymer.

**Figure 2 polymers-15-03803-f002:**
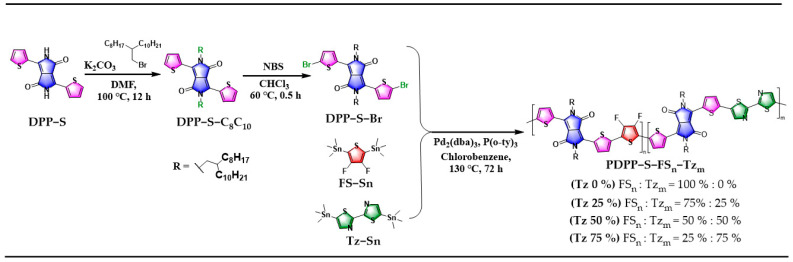
Synthesis of DPP-based monomer and PDPP−S−FS_n_−Tz_m_ polymers: Tz 0%, Tz 25%, Tz 50%, Tz 75%.

**Figure 3 polymers-15-03803-f003:**
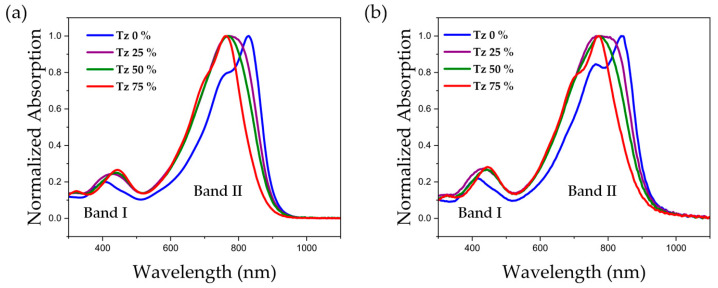
Normalized UV–vis spectra of the four polymers in (**a**) chlorobenzene solutions and (**b**) annealed thin films.

**Figure 4 polymers-15-03803-f004:**
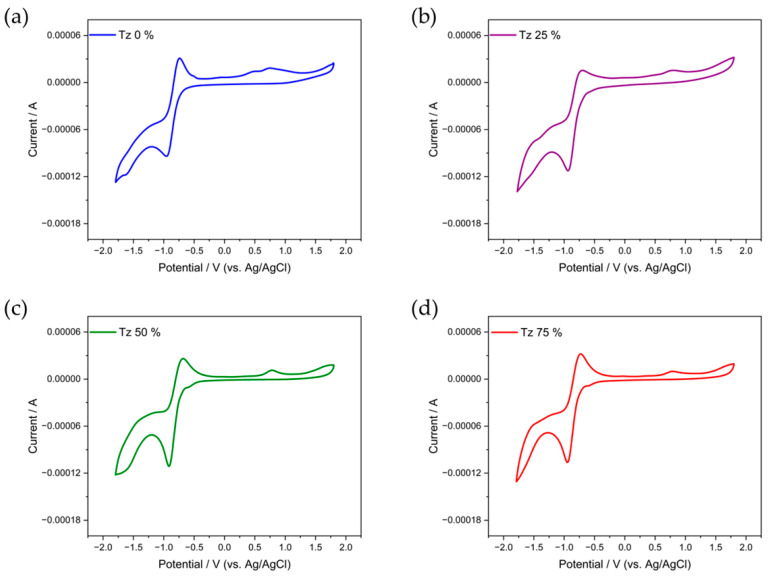
CV of (**a**) Tz 0%-, (**b**) Tz 25%-, (**c**) Tz 50%-, and (**d**) Tz 75%-based films in the acetonitrile solution with positive sweeps.

**Figure 5 polymers-15-03803-f005:**
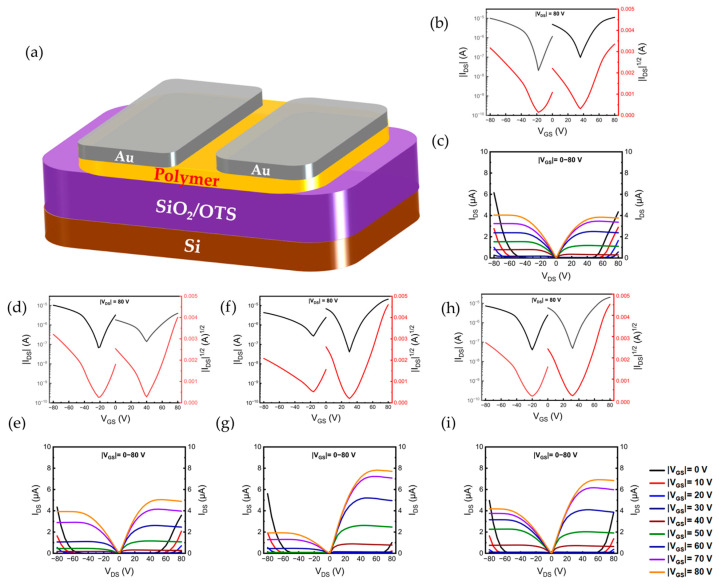
(**a**) OTFT devices with BGTC architectures. (**b**,**d**,**f**,**h**) Current–voltage characteristics of OTFTs measured under vacuum: Tz 0%, Tz 25%, Tz 50%, and Tz 75%, respectively. (**c**,**e**,**g**,**i**) Output characteristics of Tz 0%-, Tz 25%-, Tz 50%-, and Tz 75%-based devices, respectively (Different colored lines represent the gate voltage, with the left side showing a negative voltage and the right side a positive voltage).

**Figure 6 polymers-15-03803-f006:**
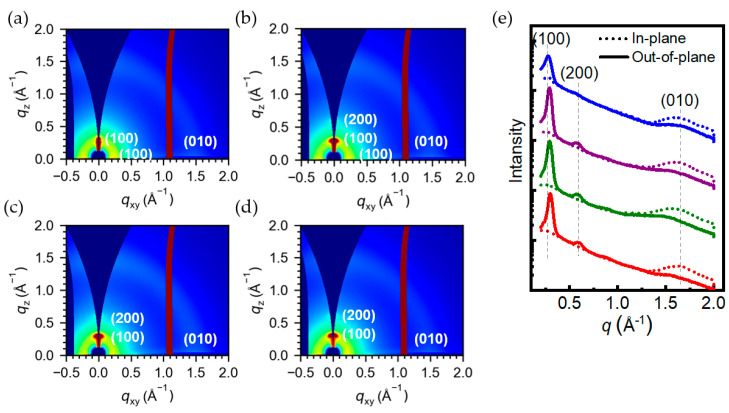
The 2D-GIWAXS patterns of (**a**) Tz 0%, (**b**) Tz 25%, (**c**) Tz 50%, and (**d**) Tz 75%, and (**e**) 1D-GIWAXS films annealed at 200 °C (blue: Tz 0%; purple: Tz 25%; green: Tz 50%; red: Tz 75%).

**Table 1 polymers-15-03803-t001:** The results of molecular weight and elemental analyses of the four polymers.

	Mn	Mw	PDI	C ^1^	H ^1^	N ^1^	S ^1^
				(%)	(%)	(%)	(%)
Tz 0%	24,541	85,717	3.49	71.20	9.10	2.65	9.90
Tz 25%	21,933	96,864	4.41	70.85	9.01	3.41	10.69
Tz 50%	28,506	139,502	4.89	70.73	8.85	4.04	11.08
Tz 75%	24,390	107,112	4.39	70.19	8.82	4.57	11.83

^1^ Based on the average of two tests.

**Table 2 polymers-15-03803-t002:** UV–vis absorption of the four polymers.

	λ_max_ (nm) ^1^	λ_max_ (nm) ^1^	λ_onset_ (nm) ^1^	λ_max_ (nm) ^2^	λ_max_ (nm) ^2^	λ_onset_ (nm) ^2^	E_g_^soln^ (eV) ^3^	E_g_^film^ (eV) ^4^
Tz 0%	829	406	906	841	421	925	1.34	1.31
Tz 25%	775	427	899	782	433	921	1.35	1.32
Tz 50%	769	441	895	775	442	914	1.36	1.33
Tz 75%	762	444	878	772	446	900	1.38	1.36

^1^ In solution. ^2^ In film. ^3^ Calculations come from 1240/λ_onset_ in solution. ^4^ Calculations come from 1240/λ_onset_ in film.

**Table 3 polymers-15-03803-t003:** Electrochemical characteristics of the four polymers.

	E_red_ (V)	E_red_^onset^ (V)	LUMO (eV) ^5^	E_ox_ (V)	E_ox_^onset^ (V)	HOMO (eV) ^6^	E_g_ ^cv^ (eV) ^7^
Tz 0%	−0.96	−0.75	−3.66	0.49	0.29	−4.70	1.04
Tz 25%	−0.94	−0.72	−3.69	0.79	0.58	−4.99	1.30
Tz 50%	−0.91	−0.71	−3.70	0.79	0.60	−5.01	1.31
Tz 75%	−0.94	−0.76	−3.65	0.78	0.62	−5.03	1.38

^5^ E_LUMO_ = −4.80 eV − [(E_red_^onset^) − E_1/2_(ferrocene)]. ^6^ E_HOMO_ = −4.80 eV − [(E_ox_^onset^) − E_1/2_(ferrocene)]. ^7^ E_g_^cv^ = E_HOMO_ − E_LUMO_.

**Table 4 polymers-15-03803-t004:** Electron and hole transport properties of PDPP−S−FS_n_−Tz_m_-based OTFT devices.

	Coating Speed (mm/s)	Annealing (°C)	Max Electron Mobilities (cm^2^/(V s))	Electron Mobilities ^1^ (cm^2^/(V s))	Max Hole Mobilities ^1^ (cm^2^/(V s))	Hole Mobilities ^1^ (cm^2^/(V s))
Tz 0%	2000	200	0.38	0.25 ± 0.11	0.070	0.053 ± 0.013
Tz 25%	2000	200	0.48	0.36 ± 0.08	0.064	0.050 ± 0.011
Tz 50%	2000	200	0.69	0.50 ± 0.08	0.036	0.022 ± 0.014
Tz 75%	2000	200	0.57	0.42 ± 0.10	0.051	0.035 ± 0.016

^1^ Based on eight OTFT devices for each condition.

**Table 5 polymers-15-03803-t005:** Crystallographic information of the PDPP−S−FS_n_−Tz_m_ series of polymers.

	In-Plane (010) Peak Position (Å^−1^)	In-Plane (010) π-Spacing (Å)	In-Plane (100) Peak Position (Å^−1^)	In-Plane (100) d-Spacing (Å)	Out-of-Plane (100) Peak Position (Å^−1^)	Out-of-Plane (100) d-Spacing (Å)
Tz 0%	1.58	3.98	0.27	23.26	0.28	22.43
Tz 25%	1.58	3.98	0.26	24.15	0.29	21.65
Tz 50%	1.59	3.95	−	−	0.30	20.93
Tz 75%	1.58	3.97	−	−	0.30	20.93

**Table 6 polymers-15-03803-t006:** Typical examples of polymer-based organic thin-film transistors in the last five years.

Materials	Electron Mobilities (cm^2^/(V s)	Hole Mobilities (cm^2^/(V s)	Device Structure	Average Electron Mobilities (cm^2^/(V s)
DPP-T	−	0.094	BGBC	−
DPP-FDT	0.51	0.80	BGBC	0.50
DPP-F_1_Ph	0.26	0.20	BGTC	0.30
DPPTh-4ClBT	0.82	0.73	TGBC	−
F2CNTVT-DPP	2.03	−	TGBC	1.50
P2DPP-2FBTz	0.68	−	TGBC	0.61
DPP-2T-DPP-TBT	3.84	3.01	TGBC	3.00
PFIDTO-T	0.27	−	TGBC	0.03
PBN-27	0.34	−	TGBC	0.32
PNFFN-DTE	1.82	−	TGBC	1.30
PNFFN-FDTE	10.74	−	TGBC	10.56
PDTzNTI	1.22	−	TGBC	0.87

## Data Availability

Not applicable.

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
