# Peer review of "Rational Design of Novel Conjugated Terpolymers Based on Diketopyrrolopyrrole and Their Applications to Organic Thin-Film Transistors"

_polymers, 2023, doi:10.3390/polym15183803_

Round 1

Reviewer 1 Report

In this paper, the Authors described the synthesis of a terpolymer and its application in Organic Thin film transistors. All the reserach presented by  the Authors  has been well thought out and was properly executed.

However, the Authors should include in the publication studies confirming the structure of the obtained polymers. I understand that those polymers have poor solubility in organic solvents (Even though the authors included information that the obtained compounds are soluble in chloroform) , but Authors should provide both NMR (e.g. solid state) and IR spectra. 

Also Figure 5 is missing the Y axis scale. This must be added. 

I reccomend publishing this paper after major revision, as confirmation of the structure of the synthesized product is an important part of this work.

Reviewer 2 Report

The authors have presented conjugated Terpolymers and their OTFT applications. Manuscript was well constructed and presented especially on the structure property relationship towards high carrier mobility. UV-Visible spectroscopy, CV and OTFT performance of four polyerms were well placed and explained along with crystallinity analysis. However, its in-deed to compare with the current literatures for device performance analysis as a form of table showing the best of your novel conjugated polymers. This will increase the readability of your manuscript. Insisting to include the device performance analysis with other literatures in table no.4. 

Authors english language sounds well. 

Reviewer 3 Report

see document

N/A

Round 2

Reviewer 1 Report

The authors presented the NMR spectra, which is necessary for the confirmation of synthesized products, and provide answers to my previous comments. The manuscript can be published in its current form

Reviewer 3 Report

I would encourage the authors to add some of the reproducibility tests of the electrochemical determination of the band structure in the SI. It would make the claims based on Figure 4 stronger.